# Somatic Mutational Landscape in Mexican Patients: *CDH1* Mutations and chr20q13.33 Amplifications Are Associated with Diffuse-Type Gastric Adenocarcinoma

**DOI:** 10.3390/ijms231911116

**Published:** 2022-09-21

**Authors:** Dennis Cerrato-Izaguirre, Yolanda I. Chirino, Diddier Prada, Ericka Marel Quezada-Maldonado, Luis A Herrera, Angélica Hernández-Guerrero, Juan Octavio Alonso-Larraga, Roberto Herrera-Goepfert, Luis F. Oñate-Ocaña, David Cantú-de-León, Abelardo Meneses-García, Patricia Basurto-Lozada, Carla Daniela Robles-Espinoza, Javier Camacho, Claudia M. García-Cuellar, Yesennia Sánchez-Pérez

**Affiliations:** 1Departamento de Farmacología, Centro de Investigación y de Estudios Avanzados del I.P.N. (CINVESTAV), Avenida Instituto Politécnico Nacional No. 2508, Ciudad de México CP. 07360, Mexico; 2Subdirección de Investigación Básica, Instituto Nacional de Cancerología (INCan), San Fernando No. 22, Tlalpan, Ciudad de México CP. 14080, Mexico,; 3Unidad de Biomedicina, Facultad de Estudios Superiores Iztacala, Universidad Nacional Autónoma de México, Los Reyes Iztacala, Tlalnepantla de Baz, Estado de México CP. 54090, Mexico; 4Instituto Nacional de Medicina Genómica (INMEGEN), Periférico Sur No. 4809, Arenal Tepepan, Tlalpan, Ciudad de México CP. 14610, Mexico; 5Servicio de Endoscopía, Instituto Nacional de Cancerología (INCan), San Fernando No. 22, Tlalpan, Ciudad de México CP. 14080, Mexico; 6Departamento de Patología, Instituto Nacional de Cancerología (INCan), San Fernando No. 22, Tlalpan, Ciudad de México CP. 14080, Mexico; 7Subdirección de Investigación Clínica, Instituto Nacional de Cancerología (INCan), San Fernando No. 22, Tlalpan, Ciudad de México CP. 14080, Mexico; 8Dirección de Investigación, Instituto Nacional de Cancerología (INCan), San Fernando No. 22, Tlalpan, Ciudad de México CP. 14080, Mexico; 9Dirección General, Instituto Nacional de Cancerología (INCan), San Fernando No. 22, Tlalpan, Ciudad de México CP. 14080, Mexico; 10Laboratorio Internacional de Investigación Sobre el Genoma Humano, Universidad Nacional Autónoma de México, Santiago de Querétaro CP. 76010, Mexico; 11Wellcome Sanger Institute, Hinxton, Cambridgeshire CB10 1SA, UK

**Keywords:** gastric cancer, Hispanic ethnicity, cancer genomics, precision medicine, copy number variation, mutations

## Abstract

The Hispanic population, compared with other ethnic groups, presents a more aggressive gastric cancer phenotype with higher frequency of diffuse-type gastric adenocarcinoma (GA); this could be related to the mutational landscape of GA in these patients. Using whole-exome sequencing, we sought to present the mutational landscape of GA from 50 Mexican patients who were treated at The Instituto Nacional de Cancerología from 2019 to 2020. We performed a comprehensive statistical analysis to explore the relationship of the genomic variants and clinical data such as tumor histology and presence of signet-ring cell, *H. pylori*, and EBV. We describe a potentially different mutational landscape between diffuse and intestinal GA in Mexican patients. Patients with intestinal-type GA tended to present a higher frequency of *NOTCH1* mutations, copy number gains in cytobands 13.14, 10q23.33, and 12q25.1, and copy number losses in cytobands 7p12, 14q24.2, and 11q13.1; whereas patients with diffuse-type GA tended to present a high frequency of *CDH1* mutations and CNV gains in cytobands 20q13.33 and 22q11.21. This is the first description of a mutational landscape of GA in Mexican patients to better understand tumorigenesis in Hispanic patients and lay the groundwork for discovering potential biomarkers and therapeutic targets.

## 1. Introduction

Worldwide, gastric cancer is the fifth cause of cancer-related deaths, with a 5-year survival rate of less than 30% in Western countries [1,2]. Asia has the highest incidence of gastric cancer, followed by Latin America (LATAM) and Europe [3]. Although LATAM ranks second in incidence, the knowledge regarding the molecular landscape of gastric cancer in this region is scarce [4]. Poor dietary habits, tobacco usage, and pathogens such as *Helicobacter pylori* and Epstein–Barr virus (EBV) are risk factors for gastric cancer [5]. Up to 95% of all patients with gastric cancer are diagnosed as having gastric adenocarcinoma (GA). According to the Lauren classification, the tumor histology of GA is classified as diffuse, intestinal, or mixed type [6]. The intestinal-type GA has been reported as the common type across Asian and European-descent ethnicities; however, a higher proportion of diffuse-type GA has been reported among the Hispanic population [7,8]; thus, a better comprehension of the molecular landscape of GA could help to understand carcinogenesis and develop new therapeutic approaches in this population.

A differential mutational pattern has been reported between diffuse- and intestinal-type GA. Somatic variations in *RHOA*, *CDH1*, *TAF1*, and *KMT2C* have been associated with diffuse-type GA [9,10]. Gene fusions, such as *CLDN18*-*ARHGAP26/6*, deletions in 9p, and gains in 20q are also some structural variations related to diffuse-type GA and signet-ring cell formations [9,11]. In contrast, intestinal-type GA is characterized by a high frequency of copy number variations (CNVs), commonly leading to the activation of tyrosine kinase-associated receptors, including ERBB2 and EGFR [12]. The study of GA mutational patterns has led to the identification of biomarkers and the use of targeted therapies, including those with trastuzumab and pembrolizumab, that benefit mainly patients with intestinal-type GA; on the other hand, diffuse-type GA has been associated with low benefit from chemotherapy [13].

The Cancer Genome Atlas (TCGA) consortium has divided GA into four molecular subtypes: EBV-positive (EBV), microsatellite-instable (MSI), chromosome-instable (CIN), and genome-stable (GS). Each GA subtype displays unique molecular and clinical characteristics [14]. The ethnicity of patients has an impact on the distribution of these subtypes; for instance, MSI and EBV subtypes are more common in South Korean patients, while GS is more abundant in Ukrainian patients [15]. In addition, the CIN subtype is more frequent in European-descent patients than in Asian patients [16]. When identifying the clinical significance of this classification, the EBV subtype is associated with the best prognosis, and the GS is associated with the worst prognosis [17].

The Hispanic population in the United States has been associated with an early age of onset, higher frequency of diffuse-type GA, and higher risk of peritoneal carcinomatosis compared with European-descent, Asian, and African American patients [18,19,20,21]. This characteristic has been associated with a distinctive genomic profile and a high frequency of *CDH1* germline mutations [22]. In Mexico, an estimated 8804 new cases and 6735 deaths related to gastric cancer were recorded in 2020 [3]. Differences in GA incidence are reported within the country, with the south and central regions especially affected [23]. The Mexican population has a rich genetic admixture with traces of Amerindian, European, African, and Asian ancestry, which varies across the country [24,25]. This genetic peculiarity may be related to the evolution of GA and the high frequency of diffuse-type GA. The GA mutational landscape has been described in populations of European-descent and Asian ethnicities; however, mutational data from the Hispanic population are elusive. Here, we sought to identify the somatic variants, tumor mutational burden (TMB), affected signaling pathways, and the CNVs that are present in Mexican patients with GA to describe a mutational landscape and explore its relationship with tumor histology and clinical characteristics.

## 2. Results

### 2.1. Clinicopathological Characteristics of the Patients

GA samples from 50 Mexican patients who were treated at The Instituto Nacional de Cancerología of Mexico (INCan) from January 2019 to January 2020 were analyzed. The mean age of patients was 54.84 years (range, 22–89 years), with a predominant percentage of males (58%). In nearly half of the patients (42%), the tumor was found to have spread within two or more sites of the stomach. Tumors located in the fundus and body were the second most common (36%), followed by tumors located in the antrum and pylorus (22%). Diffuse-type GA was detected in 62% of the samples and intestinal-type GA was detected in 32%, whereas mixed-type GA was detected in the remaining 6% of the samples. Signet-ring cells were present in 68% of patients, and poorly differentiated tumor grade was found in 84% of patients. Patients with advanced clinical stages represented 62% of the cohort. The clinical characteristics of the patients included are presented in Table 1. After an early follow-up of 2 years, we observed that patients had a median overall survival of 11 months (Appendix A). However, when comparing overall survival between patients with diffuse- and intestinal-type GA, no differences were observed (Appendix A).

### 2.2. Helicobacter Pylori and EBV Identification

*Helicobacter pylori* infection was identified in 42% of patients, whereas EBV infection was present in 12% of patients. Concomitant infection between both pathogens was present in only 8% of patients. The presence of *H. pylori* (*p*-value = 0.69) or EBV (*p*-value = 0.673) was not associated with tumor histology. The presence of signet-ring cell was also not associated with the presence of *H. pylori* (*p*-value = 0.40) and EBV (*p*-value = 0.15).

### 2.3. Tumor Mutational Burden and Clinical Features

Whole-exome sequencing (WES) was performed successfully, with one sample failing the sequencing process. An average sequencing depth of 186.90× and a minimum coverage of 99.5% of the target region were achieved. The samples had a mean of 94.86% Q30 quality in the sequenced bases. FASTQ files were obtained, and further bioinformatic analyses were performed to detect somatic variations and CNVs (see Material and Methods section). A total of 119,744 somatic variants were identified across all samples. Histograms of the variant allele fraction (VAF) for each sample are depicted in Appendix A. The total number of mutations, including driver and passenger somatic variants, present in the exome region were used to identify the tumor mutational burden (TMB) of each patient. The TMB values ranged from 0.16 to 158.35 mutations/Mb.

Patients were grouped according to their TMB status, and high TMB (TMB ≥ 10 mutations/Mb) was observed in 10% of the patients (Figure 1a). When exploring deeper into the patients with high TMB status, we observed that mutations in DNA mismatch repair were only found in patient CGTE_14 with a driver exonic mutation in *MSH6* (TV716-717X, R1024Q, T1085TX, and Q1314X), and the rest of the patients with high TMB had passenger intronic mutations in *MSH6* (CGTE_26, CGTE_18, and CGTE_40), *MSH2* (CGTE_40 and CGTE_28) and *MLH1* (CGTE_18). The potential relationship between the TMB values and clinical features was explored. Patients with intestinal-type GA (*p*-value = 0.026) and absence of signet-ring cell (*p*-value = 0.015) had greater TMB values (Figure 1b,c). The presence of *H. pylori* (*p*-value = 0.54) and EBV infection (*p*-value = 0.87) were not associated with TMB values (Appendix A). We also evaluated if the TMB status of the patients affected their overall survival (Appendix A); however, within this early follow-up time, no differences were identified between the groups (*p*-value = 0.78).

### 2.4. Somatic Variants and Tumor Histology

Somatic variants were classified as driver and passenger using the BosstDM algorithm (*see* materials and methods). We found that eight samples had missing driver mutations. The tumor suppressor gene *TP53* was the most frequently mutated gene, altered in 41% of the patients. Missense variants were the most common mutation type observed across the genes (Figure 2).

Patients were sorted by tumor histology, and the frequency of mutated genes between diffuse-type and intestinal-type GA was assessed. *NOTCH1* somatic variants were found exclusively in patients with intestinal-type GA histology (*p*-value = 0.03, false discovery rate (FDR) = 0.57). On the other hand, patients with diffuse histology tended to be associated with somatic variants in *CDH1* (*p*-value = 0.08, FDR = 0.57). Mutations in *RHOA* (*p*-value = 0.26, FDR = 0.57) and *CDKN2A* (*p*-value = 0.26, FDR = 0.57) were found to be present only in patients with diffuse-type GA. When comparing the presence of signet-ring cell and the frequency of somatic variants, mutations in *PIK3CA* (*p*-value < 0.01, FDR = 0.56) and *NOTCH1* (*p*-value = 0.03, FDR = 0.50) were exclusively present in patients without signet-ring cells. Somatic variants in *ARID1A* (*p*-value = 0.08, FDR = 0.56) also tended to be present in patients without signet-ring cells (Appendix A).

### 2.5. Somatic Variants, H. pylori, and EBV

Patients who were EBV-positive were associated with a higher frequency of *PIK3CA* (*p*-value < 0.01, FDR = 0.04), *ARID1A* (*p*-value < 0.01, FDR = 0.31), *SMAD4* (*p*-value = 0.01, FDR = 0.82), *BCOR* (*p*-value = 0.03, FDR = 0.87), and *RHOA* (*p*-value = 0.03, FDR = 0.87) somatic variants. No associations between the presence of *H. Pylori* infection and the frequency of somatic variants were found (Appendix A).

### 2.6. Signaling Pathways Affected by Somatic Mutations According to Tumor Histology

Patients were grouped according to their tumor histology to observe the differences between diffuse- and intestinal-type GA in the levels of enrichment ratio for the signaling pathways affected by driver mutations. Using the PANTHER dataset as reference [26], the hypoxia response via HIF activation was the most enriched signaling pathway in intestinal-type GA (enrichment ratio (ER) = 33%, FDR < 0.01) and diffuse-type GA (ER = 29.89, FDR < 0.01). Enrichment of the p53-related pathways, as well as the insulin/IGF and VEGF signaling pathways, was shared between diffuse- and intestinal-type GA. However, the JAK/STAT signaling pathway and Axon guidance mediated by netrin tended to be more enriched in diffuse-type GA (Figure 3a).

The biological processes affected by somatic variants were different in patients with diffuse- and intestinal-type GA, and Figure 3b shows the biological processes affected by somatic variants according to the Gene Ontology database. The mismatch repair mechanism (ER = 42.80, FDR < 0.01) was the most affected biological process in patients with intestinal-type GA, whereas the biosynthetic process of phosphatidylinositol (ER = 18.58, FDR < 0.01) was affected in patients with diffuse-type GA.

### 2.7. Copy Number Variations and Tumor Histology

CNVs affecting patients with GA were identified for different genomic regions. CNV gains were the most commonly found within the patients (Appendix A). Chromosomes 2, 7, 8, and 20 were affected almost entirely by CNV gains, whereas chromosomes 4 and 17 were affected by CNV losses. Even though large genomic regions seem to be affected by CNVs, our cohort showed high-density areas where CNV tends to occur more frequently (Figure 4a). Cytoband 20q13.33 was the most frequently affected, altered in 73.46% of the patients. This cytoband harbors different genes including *LAMA5*, which encodes for the alpha subunit of laminin 5. *LAMA5* was the most frequently affected gene by CNVs, CNV losses occurring in 61.11% of the cases. *SYCP2*, also present in 20q13.33, encodes for a protein from the synaptonemal complex and was affected by CNV gains in 93.75% of the cases. *NOTCH1*, harbored in cytoband 9q34.4, presented CNV losses in 71% of the cases, whereas *JAG2*, which is a *NOTCH1* ligand and is encoded within cytoband 14q32.33, was affected by CNV losses in 78.57% of the cases. A list of all the CNVs found in this study and the genes affected within the cytobands can be found in Appendix A.

When grouping the patients according to their tumor histology (Figure 4b), patients with diffuse-type GA tended to have gains in 20q13.33 (*p*-value = 0.01, FDR = 1) and 22q11.21 (*p*-value = 0.04, FDR = 1), whereas patients with intestinal-type GA presented a higher frequency of CNV losses in 7p12 (*p*-value < 0.01, FDR = 0.27), 14q24.2 (*p*-value = 0.01, FDR = 0.66), and 11q13.1 (*p*-value = 0.02, FDR = 0.66). Patients with intestinal-type GA also showed a high frequency of CNV gains in cytoband 13.q14 (*p*-value = 0.04, FDR = 1), 10q23.33 (*p*-value = 0.03, FDR = 1), and 15q25.1 (*p*-value = 0.03, FDR = 1). In addition, the presence of signet-ring cells was associated with CNV gains in cytobands 6p22.2 (*p*-value = 0.02, FDR = 0.67) and 1q21.3 (*p*-value = 0.07, FDR = 0.67). CNV losses in 4p16.33 were associated with absence of signet-ring cells (*p*-value = 0.03, FDR = 0.67) (Appendix A).

### 2.8. Copy Number Variations, H. pylori, and EBV

When grouping patients according to the presence of *H. pylori*, patients with *H. Pylori* presented a higher frequency of CNV gains in 17q21.2 (*p*-value = 0.02, FDR = 0.57), 10q26.3 (*p*-value = 0.02, FDR = 0.57), and 10q25.3 (*p*-value = 0.02, FDR = 0.57). CNV losses in 12q12 (*p*-value = 0.01, FDR = 0.57) were also common in patients with *H. pylori* infection. Compared to when grouping according to the EBV presence, we observed that patients with EBV infection tended to present a high frequency of CNV gains in cytobands 19q13.42 (*p*-value = 0.01, FDR = 087), 18q22.2 (*p*-value = 0.01, FDR = 0.67), and 2p25.1 (*p*-value = 0.01, FDR = 0.67). Differences in CNV losses were not found between the patients with and without EBV infection (Appendix A).

### 2.9. Mutational Processes and Their Associations with Clinical–Pathological Characteristics

The mutational processes involved in GA carcinogenesis of the patients were quantified through a mutational signature analysis to identify the contribution of single-base signatures (SBSs). The SBS16 related to liver cancer [27] and high mutational rate due to an inefficient nucleotide excision repair pathway [28] was the most common signature, present in 33% of the patients that showed a contribution higher than 0.2 (Appendix A), followed by the SBS1 related to the aging process, which was found to be overrepresented in 28%.

Patients were categorized according to similarities in the pattern of the SBS contributions; however, no defined clusters were found. When comparing the contribution of SBS with the clinical features of the patients, we identified that tumors with high TMB had higher contribution of SBS16 (Figure 5a), patients with signet-ring cells had a lower contribution of SBS01 (Figure 5b), and patients with intestinal-type GA displayed a higher contribution of SBS17 (Figure 5c). Appendix A shows a complete panorama of the associations between the most frequent mutational signatures (SBS01, SBS03, SBS05, SBS09, SBS16, and SBS17) found in our cohort and the clinical–pathological characteristics of the patients.

### 2.10. Comparison of Genes Affected by Somatic Variants and Copy Number Variations between Mexican Patients and Other Cohorts

To identify the similitudes and differences in the mutational landscape of Mexican patients with the mutated genes reported by European-descent and Asian patients, we compared this cohort with two other cohorts. We compared the patients included in this study (the INCan cohort) with the TCGA stomach adenocarcinoma firehouse legacy study cohort [14] and the OncoSG cohort [29]. The TCGA cohort includes European-descent and Asian patients, the OncoSG cohort includes Asian patients, and the INCan cohort includes Mexican patients. A total of 14,296 genes affected by somatic variants and 10,521 genes affected by CNVs were shared between the three cohorts (Figure 6a). However, about 5000 mutated genes were exclusive to the INCan cohort. *TP53* was the gene most frequently mutated across all cohorts. CDH1 seemed to be more frequently mutated in the INCan cohort (Figure 6b). A list showing the frequency of all the genes with driver mutations in the three cohorts can be found in Appendix A. Additionally, the genes encoded in cytoband 1p36.33: *AGRN*, *MEGF6*, *TNFRSF18*, and *TNFRSF4* seemed to be more affected by CNVs in the INCan cohort; they were affected by CNV losses. *WDR90* and *ABCA2*, which are encoded within cytobands 16p13.3 and 9q34.4, respectively, were also affected by CNV losses.

## 3. Discussion

To the best of our knowledge, this is the first study able to describe a mutational landscape of GA in a cohort of Mexican patients using fresh samples. Here, we analyzed the somatic variants and CNVs using WES and identified a mutational landscape for Mexican patients with GA. We compared a potentially different mutational landscape between diffuse- and intestinal-type GA in Mexican patients. Patients with intestinal-type GA tended to present a higher frequency of *NOTCH1* mutations, CNV gains in cytobands 13.14, 10q23.33, and 12q25.1, and CNV losses in cytobands 7p12, 14q24.2, and 11q13.1; whereas patients with diffuse-type GA tended to present a higher frequency of *CDH1* mutations and CNV gains in cytobands 20q13.33 and 22q11.21.

This cohort showed similar clinicopathological characteristics reported for other Mexican and Hispanic patients with GA. The patients in this study had a higher proportion of diffuse-type GA, clinical state III–IV, and a low proportion of EBV infection. These characteristics align with those reported in other studies using larger cohorts of Mexican patients, including a previous study by our team [30], which also mentioned a predominance of diffuse-type GA [31,32].

Similar to Asian cohorts where TMB has been proposed as a biomarker for survival and use of immunotherapy, we identified that 10% of patients with GA had high TMB [33,34]. High TMB has been associated with intestinal-type GA [35], and we found that Mexican patients with greater TMB raw values were also associated with intestinal-type GA. It seems that the Mexican patients behave similar to other cohorts in terms of TMB. We found an association between a greater contribution of SBS16 and high TMB values. However, no evidence of somatic mutations in genes associated with the mismatch repair pathway, a common explanation for high TMB of the patients with the microsatellite-instable (MSI) phenotype, was found. In African American patients with esophageal squamous cell carcinoma, a prominent association between SBS16 and high TMB was attributed to defects in the nucleotide excision repair (NER) pathway of the patients with high TMB (27,28). Other studies are necessary to expose the implications of defects on the NER pathway in Mexican patients with high TMB.

In gastric cancer, TMB does not correlate with the MSI phenotype, but both are promising predictive biomarkers for the use immunotherapy [36]. Currently, the Precision Medicine Working Group from the European Society for Medical Oncology recommends TMB as a biomarker to select PD-1/PD-L1 therapy for well- and moderately differentiated neuroendocrine tumors and cervical, salivary, thyroid, and vulvar cancers [37]; however, no recommendations have been issued for gastric cancer. The US Food and Drug Administration (FDA) has approved the use of TMB as a biomarker for selecting pembrolizumab for solid tumors with high levels of TMB (TMB > 10 mutations/mb) based on the results of the KEYNOTE-158 study [38]. In Mexico, TMB is not used as a biomarker for gastric cancer treatment. Here, we found no correlation between TMB and Lauren classification; however, other studies should be performed to investigate the use of TMB as a biomarker in other aspects of the disease, such as its utility in selecting targeted therapies.

Whole-exome sequencing studies performed in Asian and European-descent populations have identified a set of mutations and genes with the potential to drive personalized medicine in gastric cancer [35,39,40]. Mexican patients exhibited somatic variants in known cancer-related genes, such as *TP53*, *CDH1*, and *ARID1A*. In concordance with reports in the Hispanic population [22], we observed a high frequency of *CDH1* mutations. The high frequency of *CDH1* could be due to the greater number of patients with diffuse-type GA included in our cohort. *CDH1* encodes the adhesion protein E-cadherin. Somatic mutations in *CDH1* have been associated with the epithelial–mesenchymal transition in tumor cells, resulting in a more aggressive tumor phenotype [41,42]. *CDH1* mutations are commonly associated with diffuse-type GA, and here, we observed the same tendency.

Due to the high frequency of diffuse-type GA reported for Hispanic patients, we decided to focus on comparing the mutational pattern of patients with diffuse- and intestinal-type GA. Patients with diffuse-type GA display mutations in *CDH1*, *RHOA,* and *CDKN2A*. *RHOA* encodes for the small GTPase family member Rho. *RHOA* mutations have been associated with cell survival and cell migration through the inactivation of ROCK [43]. In patients with lymphoma, *RHOA* mutations were assessed as a potential biomarker for poor prognosis [44], and gain-of-function mutations were studied as a possible therapeutic target for gastric cancer [45]. All *RHOA* mutations identified in this study were missense mutations; however, due to the high frequency of diffuse-type GA in the Mexican population, further studies are required to identify their utility both as a prognosis biomarker as well as a therapeutic target in Mexican patients.

Amplifications of cytoband 20q13.33 were associated with diffuse-type gastric cancer. In colorectal cancer, amplifications in cytoband 20q13.33 were associated with early tumoral stages and mutations in *APC* and *KRAS* [46]. *LAMA5* was the most frequently affected gene within cytoband 20q13.33 in our cohort. In colorectal cancer, *LAMA5* overexpression is associated with hepatic metastasis, angiogenesis, and NOTCH1 pathway inhibition [47]. In ovarian cancer, *LAMA5* overexpression was associated with a good prognosis [48]. No information about the potential use of *LAMA5* as a biomarker for prognosis in gastric cancer has been reported. *PSMA7* and *GID8* are two genes encoded within 20q13.33 that were described as possible gene targets in gastric cancer [49]. The overexpression of *PSMA7* was associated with poor prognosis in patients with gastric cancer [50]. Identifying whether 20q13.33 is affected in the early stages of GA in Mexican patients, or which genes are affected by CNVs in this cytoband, requires further studies.

Patients with intestinal-type GA were associated with mutations in *NOTCH1.* NOTCH signaling has been widely dissected in gastric cancer tumorigenesis [51,52,53]. In colorectal cancer, increased NOTCH1 expression is associated with greater immune cell infiltration, tumor volume, and depth invasion [51,54]. In cervical cancer, the loss of nuclear NOTCH1 was reported to be an independent predictor of malignancy [55]. We identified that *NOTCH1* and *JAG2* were affected exclusively by deleterious mutations and mainly by CNV losses. We suspect that the NOTCH signaling pathway is underactive in intestinal-type GA; however, the clinical implications of these genomic variants are still to be elucidated.

A limitation of this study is the number of patients included in this cohort, which could restrict the statistical power of the analysis and hinder the frequency of genomic variants identified. We are aware that a small number of patients would restrain the representation of the true molecular landscape of gastric cancer of Mexican patients and could limit the statistical power to compare the molecular behavior of our cohort with other larger cohorts. However, in this first study using fresh samples from a prospective Mexican cohort, we reported the mutational landscape of GA and managed to identify associations between genomic variants and clinicopathological characteristics, such as tumor histology, signet-ring cell morphology, and *H. pylori* and EBV presence, which could guide us in the design of further studies that feature a larger cohort to test some of the identified variants as biomarkers of prognosis for patients treated with immune checkpoint inhibitors or other therapies.

The ethnicity of patients may influence the mutational landscape of GA. For example, it is reported that African American patients tend to present more *TP53* somatic variants than Asian, European-descent, and Hispanic patients [56]; European-descent patients tend to present deletions of the phosphatase gene *PTPRD* more frequently than Asian patients (16); and a high frequency of somatic variants of *APC*, *ARID1A*, *KMT2A*, *PIK3CA,* and *PTEN* has been found in Asian patients compared to European-descent patients [57]. Hispanics are one of the most genetically diverse ethnic groups [58,59]; however, GA somatic and structural variants in this ethnic group have not been thoroughly explored. A recent review published by our team presented the mutational landscape of Hispanic GA patients, identifying a differential distribution of the variants according to the countries [4]. Thus, Hispanics should not be considered a genetically homogeneous ethnic group in GA research. Here, we present a mutational landscape for GA in Mexican patients to better understand tumorigenesis in patients of Hispanic ethnicity and lay the groundwork for the development of potential biomarkers as well as therapeutic targets.

## 4. Materials and Methods

### 4.1. Population and Sample Collection

Patients included in this study were older than 18 years old, had a positive diagnosis of gastric adenocarcinoma, did not receive any type of treatment, and signed an informed consent prior to sample collection. Both tumor and non-tumor gastric tissue samples were collected for each patient by two trained endoscopists. The presence of tumoral cells in the tumor samples were confirmed by a trained pathologist. Samples were immediately stabilized after collection in RNAlater (Invitrogen, Waltham, MA, USA, AM7021) and stored for 24 h at 4 °C. Thereafter, RNAlater was removed, and the tissue samples were stored at −80 °C until DNA extraction.

### 4.2. Clinical Data Collection

Clinical data, including age at diagnosis, body mass index (BMI), sex, clinical stage, tumor location, endoscopic impression (Borrmann classification), tumor histology (Lauren classification), tumor grade, and the presence of signet-ring cells, were collected from the electronic clinical records of each patient.

### 4.3. DNA Extraction and Quality Control

Tumor and non-tumor samples were homogenized using QIAshredder columns (Qiagen, Germantown, MD, USA, 79654), and DNA was extracted using the AllPrep^®^ DNA/RNA Mini Kit (Qiagen, Germantown, MD, USA, 80204), following the protocol recommendations [60]. DNA purity was evaluated using NanoDrop 2000. The quantity and DNA fragmentation status were evaluated with the Agilent 2200 TapeStation System using the genomic DNA ScreenTape assay (Agilent, Santa Clara, CA, USA, 5067-5365) to obtain the DNA integrity number (DIN). Samples with a DIN between 6 and 10 proceeded to whole-exome sequencing.

### 4.4. PCR

The presence of *H. pylori* was detected in samples with 100 ng of DNA from both the tumor tissue and adjacent tissue using the Step One Plus Real-Time PCR System (Applied Biosystems, Thermo Fisher Scientific, Waltham, MA, USA). RealQ Plus 2× Master Mix (AMPLIQON, A323402) was used according to the manufacturer’s instructions. The HP-16S primer set (5′-TCGGAATCACTGGGCGTAA-3′ and 5′-TTCTATGGTTAAGCCATAGGATTTCACAC-3′) was used to detect the presence of *H. pylori.* β-Globin was used as an internal control; the specific primers used were 5′-GACAGGTACGGCTGTCATCA-3′ and 5′-TAGATGGCTCTGCCCTGACT-3′.

### 4.5. In Situ Hybridization

The presence of EBV in tumoral tissue was assessed by in situ hybridization for the EBV-encoded RNA (EBER1), following the procedures previously described [61]. A tumor was considered EBV-negative if EBER1 staining was undetected or only expressed in lymphoid cells with a benign appearance. A tumor was considered positive if EBER1 staining was observed in the nucleus of malignant cells.

### 4.6. Library Preparation, Hybridization Capture, and WES

Library preparation, hybridization capture, and WES procedures were performed by Novogene Corporation Inc. (Sacramento, California, USA). Sequencing libraries from the extracted DNA were generated using the Agilent SureSelect Human All Exon Kit V6 (Agilent, Santa Clara, California, USA, 5190-8865) following the manufacturer’s recommendations. Products were purified using the AMPure XP system (Beckman Coulter, Brea, California, USA, A63882) and quantified using an Agilent high sensitivity DNA assay on the Agilent Bioanalyzer 2100 system. The post-capture libraries were sequenced on the Illumina NovaSeq 6000 platform.

### 4.7. Bioinformatic Analysis Pipeline

Raw sequencing data were filtered to eliminate adapter-contaminated and low-quality reads. A read pair was discarded if either read contained adapter contamination, if more than 10% of bases were uncertain, or if the proportion of low-quality bases was below 50%. Burrows-Wheeler Aligner (BWA) (https://github.com/lh3/bwa accessed on 12 February 2021) (version [v] 0.7.8-r455) [62] was used to map the paired-end clean reads to the GRCh37 human reference genome. SAMtools (http://www.htslib.org accessed on 12 February 2021) (v1.0) [63] was used to sort the BAM file, and Picard (v1.111) [64] was used to mark and remove duplicated reads. Somatic variants (SNVs) were identified using three variant callers: MuTect (https://github.com/broadinstitute/mutect accessed on 12 February 2021) (v1.1.4), VarScan2 (https://github.com/Jeltje/varscan2 accessed on 12 February 2021) (v4.1) [65], and MuTect2 (https://gatk.broadinstitute.org/hc/en-us/articles/360037593851-Mutect2 accessed on 12 February 2021) (v4.2) [66]. Only the variants that passed all the filtering sets to the variant calling programs and labeled as “PASS” were selected. Only SNVs identified by two or more variant callers were selected for further analysis. Small insertions and deletions (InDels) were identified with Strelka (https://github.com/Illumina/strelka, accessed on 12 February 2021) (v1.0.13) [67], VarScan2, and MuTect2. InDels identified by two or more variant callers were selected. All the SNVs and InDels that had a minor allele frequency of 1% or higher in gnomAD (v2.0.1) were filtered out to eliminate potential germline SNP. In addition, to prevent any contamination with germline variants, we performed a paired analysis for each patient, filtering out all the mutations shared by the adjacent and the tumoral tissues and leaving only the mutations found in the tumoral tissue with a genotype 0/1 in the final VCF files; also, we filtered out variants with a variant allele frequency smaller than 0.05 and greater than 0.7. The biological and clinical relevance of SNVs was obtained using The Cancer Genome Interpreter (CGI) web interphase [68]. For predicting driver mutations, a tissue-specific model (stomach adenocarcinoma) was selected in the CGI. This model uses a machine learning algorithm named BoostDM to annotate the clinical and biological relevance of the somatic variants with a BoostDM score of 0.5 and an accuracy for predicting driver mutations (F_50_-score) above 0.9 [69]. Control-FREEC (https://github.com/BoevaLab/FREEC accessed on 12 February 2021) (v11.4) [70] was used for the identification of CNVs. Segments with log2 greater than 0.2 were classified as CNV gains, and segments with log2 lower than −0.235 were CNV losses.

### 4.8. Tumor Mutational Burden

TMB was calculated using the total variants affecting the coding region. Samples with TMB equal or greater than 10 mutations/Mb were considered as having high TMB [71].

### 4.9. Enrichment Analysis

Enrichment analysis was performed using the Gene Ontology web interphase (http://geneontology.org, accessed on 12 March 2022) with the PANTHER classification system to identify the biological processes and the molecular pathways affected by somatic variants [26]. Fisher’s exact test with the Benjamini–Hochberg procedure for *p*-value correction was used. The top-ten enriched pathways were selected.

### 4.10. Mutational Signatures

Mutational signatures’ contribution to the mutational process was calculated for each sample using the R package deconstuctSigs (v1.9.0) [72]. COSMIC signatures were set as reference signatures, and the count method was set as default. Mutational signature weight ≥0.2 was considered as a positive contribution to the mutational process of the sample.

### 4.11. TCGA and OncoSG Data Acquisition

The open-access web resource, c-Bioportal, was used to access TCGA genomic data [73,74]. Samples from the stomach adenocarcinoma (TCGA, Firehouse Legacy) study (14) and from the Gastric Cancer (OncoSG, 2018) study (29) were selected. Somatic and structural variant data were retrieved. A Venn diagram was used to observe similitudes and differences the number of genes affected by somatic variants and CNVs shared by TCGA, OncoSG, and our study. Then, the frequency of the ten most affected genes by somatic variants and CNVs in our cohort was presented in a bar plot to depict differences and similarities with the other cohorts.

### 4.12. Statistical Analysis

Associations between clinical characteristics such as: tumor histology and the presence of signet-ring cells, H. pylori, and EBV were assessed with the presence of somatic variations and CNVs. Patients were grouped according to their clinical characteristics, and then the frequency of the variants between the groups was compared using Fisher’s exact test. *p*-values were adjusted using the Benjamini–Hochberg procedure. TMB raw values were also evaluated for association with the clinical characteristics of the patients using the Kruskal–Wallis test. The positive contribution of mutational signatures was categorized and compared with the clinical characteristics of the patients. Contingency tables were analyzed for independence using the chi-squared test with Yates’ correction. Statistical significance was set at *p* < 0.05. All statistical analyses were performed using R software (https://cran.r-project.org, accessed on 28 July 2021).

## 5. Conclusions

We described a mutational landscape for Mexican patients with GA, exploring the differential mutational pattern between clinical features such as tumor histology and the presence of signet-ring cells, *H. Pylori*, and EBV infection. We focused our discussion on the potentially different mutational pattern observed between diffuse- and intestinal-type GA. Mexican patients with intestinal-type GA tended to present a high frequency of *NOTCH1* mutations, CNV gains in cytobands 13.14, 10q23.33, and 12q25.1, and CNV losses in cytobands 7p12, 14q24.2, and 11q13.1; whereas patients with diffuse-type GA tended to present a high frequency of *CDH1* mutations and CNV gains in cytobands 20q13.33 and 22q11.21. Whether these variants occur in other populations and their role in GA development and treatment require further investigation with a larger cohort.

## Figures and Tables

**Figure 1 ijms-23-11116-f001:**
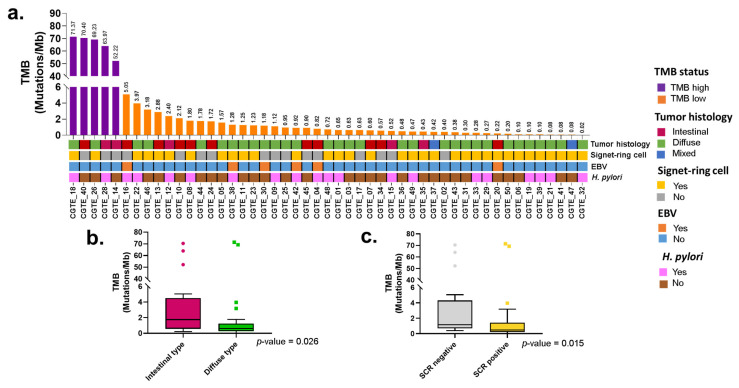
Tumor mutational burden of gastric adenocarcinoma in patients treated at The Instituto Nacional de Cancerología of Mexico between January 2019 and January 2020 (n = 49). (**a**) Tumoral mutational burden (TMB) across the sequenced samples. TMB was calculated with all the variants affecting the coding region. Purple bars indicate high TMB, and orange bars indicate low TMB. TMB ≥ 10 mutations/Mb was considered as high TMB. The lower panel represents the clinical features of the patients including tumor histology, signet-ring cell presence, Epstein–Barr virus (EBV) presence, and *Helicobacter pylori* (*H. pylori*) infection. (**b**) Comparison of TMB value according tumor histology. (**c**) Comparison of TMB value according to the presence of signet-ring cell. *p*-values were calculated using Kruskal–Wallis test.

**Figure 2 ijms-23-11116-f002:**
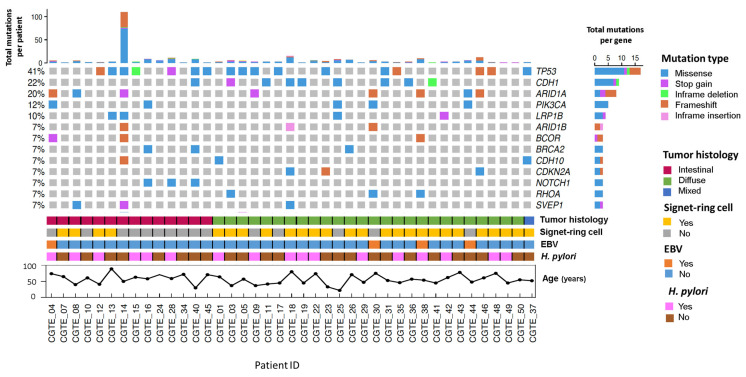
Driver somatic variants and clinical characteristics of patients with gastric adenocarcinoma treated at The Instituto Nacional de Cancerología between January 2019 and January 2020 (n = 41). Oncoprint, sorted by tumor histology, depicting the genes affected by driver somatic variants (single nucleotide variations, small insertions, and small deletions) in 7% or more of the samples. Less frequent driver somatic variants can be found in Appendix A. The variants are represented according to the mutation type, each described on the right color panel. The upper bar plot represents the number of driver somatic variants per patient, and the right bar plot represents the number of driver somatic variants per gene. The lower panner represents the tumor histology, the presence of signet-ring cells, Epstein–Barr virus (EBV), and *Helicobacter pylori* (*H. pylori*), and age per patient.

**Figure 3 ijms-23-11116-f003:**
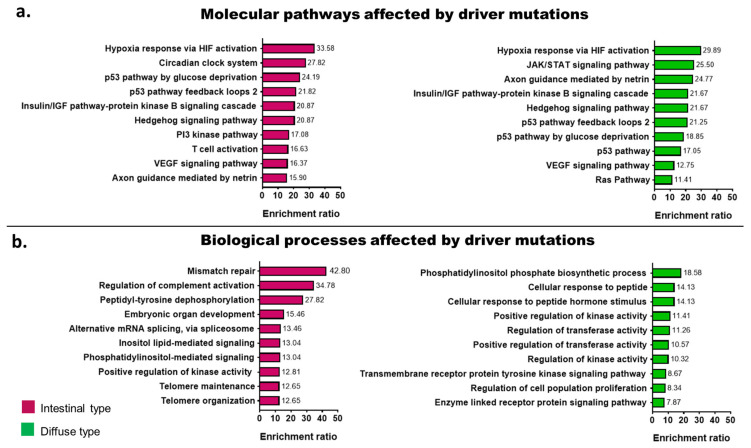
Molecular pathways affected by driver somatic variants from patients with gastric adenocarcinoma treated at The Instituto Nacional de Cancerología of Mexico between January 2019 and January 2020, grouped according to the tumor histology. (**a**) Bar plot representing the molecular pathways affected by driver somatic variants using PANTHER as reference dataset. (**b**) Bar plot representing depicting the biological processes affected by driver somatic variants, using as reference the Gene Ontology dataset.

**Figure 4 ijms-23-11116-f004:**
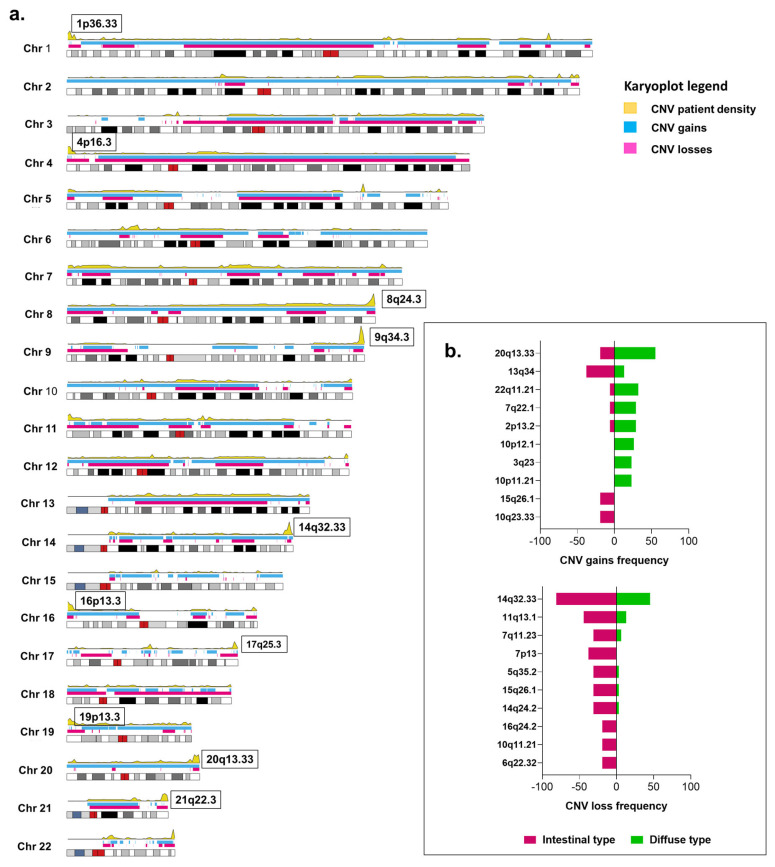
Copy number variations (CNVs) present in patients with gastric adenocarcinoma treated at The Instituto Nacional de Cancerología between January 2019 and January 2020 (n = 49). (**a**) Karyoplot depicting the genomic regions affected by CNVs in all the patients (n = 49). Blue represents the CNV gains, pink, the CNV losses. The yellow density plot represents the frequency of patients affected by CNVs across the genome. Top-ten frequently affected cytobands across all samples are shown in a box in bold. (**b**) Divergent bar plot showing the comparison between the affected cytobands in patients with diffuse- and intestinal-type gastric cancer.

**Figure 5 ijms-23-11116-f005:**
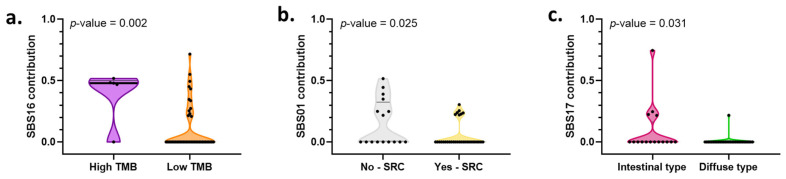
Mutational processes associated with clinical–pathological characteristics of patients with gastric adenocarcinoma treated at The Instituto Nacional de Cancerología between January 2019 and January 2020 (n = 49). (**a**) Violin plot depicting the association between the contribution of single-base substitution signature (SBS) 16 and the tumoral mutational burden (TMB) status of the patients. (**b**) Violin plot depicting the association between the contribution of SBS01 and the presence of signet-ring cells (SRSs). (**c**) Violin plot depicting the association between the contribution of SBS17 and the tumor histology.

**Figure 6 ijms-23-11116-f006:**
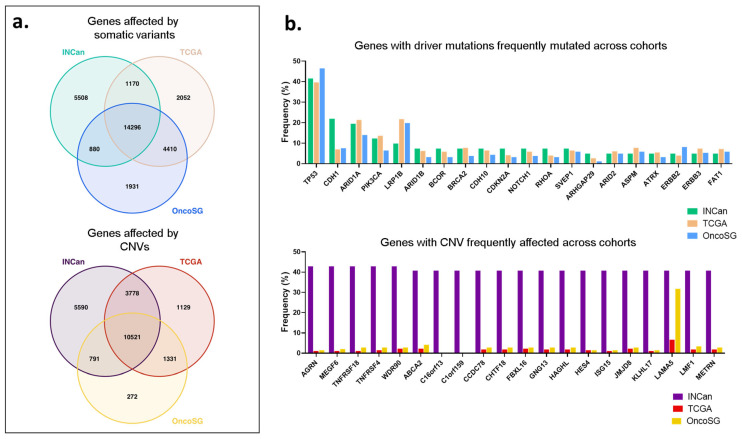
Genes affected by somatic variants and copy number variations in patients with gastric adenocarcinoma among three different cohorts. (**a**) Venn diagram depicting the number of somatic and structural variants shared between the patients from The Instituto Nacional de Cancerología (INCan), The Cancer Genome Atlas (TCGA) stomach adenocarcinoma firehouse legacy, and the OncoSG cohorts. (**b**) Bar plot presenting the 20 most frequently affected genes by driver mutations and copy number variations (CNVs) found in our analysis (INCan cohort) and their frequency found in the TCGA and OncoSG cohorts.

**Table 1 ijms-23-11116-t001:** The clinical and pathological characteristics of patients treated at The Instituto Nacional de Cancerología between January 2019 and January 2020 (N = 50).

Variable	Mean	SD
**Age** (years)	54.94	15.49
**BMI** (Kg/m^2^)	23.60	5.18
**Follow-up** (months)	12.53	14.17
**Variable**	n	%
**Sex**		
Female	21	42.00
**Clinical stage**		
I–II	3	6.00
III−IV	47	94.00
**Tumor location**		
Antrum and pylorus	11	22.00
Cardia	0	0.00
Fundus and body	18	36.00
Two or more locations	21	42.00
**Endoscopic impression ^1^**		
I	1	2.00
II	3	6.00
III	15	30.00
IV	22	44.00
V	6	12.00
Not reported	3	6.00
**Tumor histology ^2^**		
Intestinal	16	32.00
Diffuse	31	62.00
Mixed	3	6.00
**Tumor grade**		
Well-differentiated	0	0.00
Moderately differentiated	8	16.00
Poorly differentiated	42	84.00
**Signet-ring cell**		
Positive	34	68.00
** *H. pylori* **		
Positive	21	42.00
**EBV**		
Positive	6	12.00

SD: standard deviation; BMI: body mass index, EBV: Epstein–Barr virus. ^1^ Endoscopic impression is according to Borrmann classification. ^2^ Tumor histology is grouped according to the Lauren classification.

## Data Availability

The data presented in this study are available upon request to the corresponding author if you want to partner or contribute to the project. The data are not publicly available due to institutional review board politics.

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
