# Peer review of "Somatic Mutational Landscape in Mexican Patients: CDH1 Mutations and chr20q13.33 Amplifications Are Associated with Diffuse-Type Gastric Adenocarcinoma"

_ijms, 2022, doi:10.3390/ijms231911116_

Round 1

Reviewer 1 Report

The manuscript by Cerrato-Izaguirre et al. describe the somatic mutations in 50 Mexico gastric adenocarcinoma (GA) patients. The study used statistics and bioinformatics methods to show the differences between diffuse and intestinal GA on mutational landscape.

Comments:

1.     The description should be more focused on the topic. For example, the figures of the manuscript should be presented in the way to support the conclusion of the study.

2.     Figure 1, 5 didn’t describe relationship between high TMB and clinical features. As such, the significance can’t be deduced by these figures.

3.     Figure 2, need to give explanation for why in some patients “Total mutations per patient” had more mutation type (more color) than the mutation type in the column below. 

4.     “Somatic variants were classified as driver and passenger using the BoostDM 154 algorithm” - You clarified that BoostDM was used to identify driver and passenger variants, what threshold of BoostDM score was used to distinguish the driver from passenger variants, and what is the theoretical basis for the setting of threshold?

5.     I would suggest moving Figure 5 as a supplementary figure. Clustering didn’t give statistical significance and doesn't provide very useful information.

6.     Figure 6b, “Bar plot presenting the 20 most frequently…” – 20 most frequently affected genes from which gene list?

7.     Figure 6b, the bar plot showed the frequencies of sample with driver mutation affected genes. For INCan cohort, there were 49 samples, but for TCGA and OncoSG, they had hundreds of samples. The difference makes it not reliable to compare their frequencies.

8.     Line 154, a typo, should be “BoostDM”.

Reviewer 2 Report

The paper evaluated the mutational landscape of GA from 50 Mexican patients using NGS. The work is well-written and of real interest.

I would suggest to discuss more the different mutational lanscape: hispanic, asian, western-european. Maybe a table or a paragraph in the discussion may be of interest.

I think it would be interesting to discuss future purposes of your research. Are you planning to increase the patient sample? Apply your preclinical research to clinic?

Reviewer 3 Report

In this study, the authors studied the genomic alterations of gastric adenocarcinoma from 50 Mexican patients and compared among different histology types. The sequencing data can be a useful resource to researchers studying gastric adenocarcinoma. However, much of the analyses seem not very carefully performed and the related results appear questionable. A number of questions need to be addressed.

1.     Five samples have exceedingly high mutation burden (>65 mutations /Mb). This is not a frequently observed phenomenon. The authors definitely need to look deeper into this and provide explanation or interpretation. Do they have microsatellite instability? If so, which of the DNA mismatch repair gene(s) is mutated or deleted? Or did they receive any chemotherapy? What are the distributions of mutation allele frequency of the mutations in these samples?

2.     Related to point 1. Could the hypermutation reflect less rigorous mutation filtering process? From the method section the authors required at least 2 tools to support mutation or indel, this is OK, but further filtering on the coverage and number of mutated reads is probably still needed. Also did the authors remove potential germline SNPs, especially those with high frequency in human populations (e.g. 1%)? Samples CGTE_26 and 40, for example, showed the highest mutation burden. However, the major signatures are aging/clock like signatures SBS1/SBS5 (Figure 5). It is hardly imaginable that these signatures can produce such high number of mutations. SBS11 (temozolomide related chemotherapy signature) was somehow omitted in Figure 5. Figure 1 and Figure 5 should also be merged as two different panels of the same figure.

3.     Figure 6a. Most of the somatic mutations in this cohort exactly match those observed in TCGA and OncoSG cohorts? This seems odd. Unrelated tumors usually share very few somatic mutations if they are not hotspot mutations from oncogenes.

4.     Are the mutations provided in Figure 2 all pathogenetic or likely pathogenetic mutations?  Two samples with ARID1B mutations also showed concurrent ARID1A mutations. What are these mutations in these two samples? Are they really somatic and pathogenetic?

5.     Figure 6b. TTN is the largest gene and thus mutated frequently. However, it is not an oncogene or tumor suppressor gene, and should be removed. Please also check other genes in the list.

6.     When comparing different histological conditions, the authors occasionally reported FDR, but mostly did not. They should be consistent and report FDR whenever there are multiple comparisons.

7.     The authors mentioned structural variants but actually performed CNV analysis. Please replace all structural variants with copy number variations.

8.     What are the purity and ploidy of the samples? How did the authors define copy number gain and loss?

Round 2

Reviewer 1 Report

The revision satisfactorily answers each comment for the previous submission. 

Author Response

We apreciate the comments of Reviewer 1

Reviewer 3 Report

Basically, the authors revised manuscript did not provide enough insight regarding the cause of the ultra-high mutation burden of the 5 samples. If there is a reliable explanation, I believe it would be a highlight of this work. However, at the moment I am concerned that most of these mutations may not be really somatic.

The mismatch repair system does not seem much related since only exonic mutation was observed in MSH6 in one patient. The statement “these patients presented mutations in genes related to the mismatch repair system” is actually misleading. The authors should spell out the specific amino acid changes of MSH6 and other driver mutations that they claimed when possible.

They filtered mutations with VAF high than 0.7. This is not reasonable, since for tumors with very high purity, clonal mutations at copy loss LOH regions will be close to 1.

For an overall diploid tumor genome (not hyper-diploid ones), VAF usually is peaked at around half the purity. Many samples showed peak lower than 0.1 (such as sample 07, 11, 17, 22 and others). This raises the question that some of the sample may be with very low purity (i.e. <=20%).

The authors showed VAF for 4 of the 5 hyper mutation samples (14, 26, 28 and 40). Why case 18 was not shown? For cases 26, 28 and 40, the VAF all peaks at around 0.5, indicating either that the purities were around 1 or that most of the mutations were germline.

After running Mutect2, there is a filtering step which will add labels for each mutation. Did the authors only take the ones labeled as “PASS”?

The authors should have tumor purity and ploidy estimated. It will give a much clear idea about the quality of the sequencing data and the reliability of the mutations.

The copy number segmentations and the Log2 ratio of all samples should also be provided.

For mutations in Supplementary Table 1, VAFs should also be provided.

What is the point of Figure 5A? Nothing can be learned from the Venn diagram.

“Structural variants” was still kept in the name of a few sections.

Round 3

Reviewer 3 Report

I do not have additional comments for the authors